# Emulative, coherent, and causal dynamics between large-scale brain networks are neurobiomarkers of Accelerated Cognitive Ageing in epilepsy

Antoine Bernas[1,2]*, Lisanne E. M. Breuer[3], Albert P. Aldenkamp[1,2,3], Svitlana Zinger[1]

**1** Department of Electrical Engineering, Eindhoven University of Technology, Eindhoven, The Netherlands, **2** Department of Cognitive Neuropsychiatry and Clinical Neurosciences, School for Mental Health and Neuroscience, Maastricht University, Maastricht, The Netherlands, **3** Department of Research and Development, Epilepsy Centre Kempenhaeghe, Heeze, The Netherlands

* antoine.bernas1@gmail.com

**Data Availability Statement:** All relevant data are within the manuscript and its Supporting Information files. The raw clinical data cannot be shared publicly because of privacy reasons and the

## Abstract

Accelerated cognitive ageing (ACA) is an ageing co-morbidity in epilepsy that is diagnosed through the observation of an evident IQ decline of more than 1 standard deviation (15 points) around the age of 50 years old. To understand the mechanism of action of this pathology, we assessed brain dynamics with the use of resting-state fMRI data. In this paper, we present novel and promising methods to extract brain dynamics between large-scale resting-state networks: the emulative power, wavelet coherence, and granger causality between the networks were extracted in two resting-state sessions of 24 participants (10 ACA, 14 controls). We also calculated the widely used static functional connectivity to compare the methods. To find the best biomarkers of ACA, and have a better understanding of this epilepsy co-morbidity we compared the aforementioned between-network neurodynamics using classifiers and known machine learning algorithms; and assessed their performance. Results show that features based on the evolutionary game theory on networks approach, the emulative powers, are the best descriptors of the co-morbidity, using dynamics associated with the default mode and dorsal attention networks. With these dynamic markers, linear discriminant analysis could identify ACA patients at 82.9% accuracy. Using wavelet coherence features with decision-tree algorithm, and static functional connectivity features with support vector machine, ACA could be identified at 77.1% and 77.9% accuracy respectively. Granger causality fell short of being a relevant biomarker with best classifiers having an average accuracy of 67.9%. Combining the features based on the game theory, wavelet coherence, Granger-causality, and static functional connectivity- approaches increased the classification performance up to 90.0% average accuracy using support vector machine with a peak accuracy of 95.8%. The dynamics of the networks that lead to the best classifier performances are known to be challenged in elderly. Since our groups were age-matched, the results are in line with the idea of ACA patients having an accelerated cognitive decline. This classification pipeline is promising and could help to diagnose other neuropsychiatric disorders, and contribute to the field of psychoradiology.

General Data Protection Regulation (GDPR) compliance.

**Funding:** The authors received no specific funding for this work.

**Competing interests:** The authors have declared that no competing interests exist.

# 1 Introduction

Cognitive impairment is a common comorbidity in epilepsy. It is estimated that 65% of all patients show impairment in cognition, accounting for about half of the burden of disease [1]. Using fMRI, research has mainly focused on specific cognitive impairments, and correlated them with aberrant functional connectivity in brain areas involved in focal epilepsy. For instance, in rolandic epilepsy, language and motor-related functions are impaired, and were linked to a reduced functional connectivity between sensorimotor network and Broca's area [2]. Also already at young age, children with frontal lobe epilepsy complicated by cognitive impairment (memory) showed decrease in frontal lobe connectivity [3]. In an elderly population, the decline in cognitive abilities is also clearly present, yet often neglected [4]. This more general cognitive decline is often due to long-term accumulating effects, i.e. early onset and chronic epilepsy, and heavy use of anti-epileptic drugs, which hinders healthy mental ageing [5]. This is often described as a chronic/accumulative model of cognitive decline, and viewed as a form of dementia [6]. More recently, another model of cognitive decline in elderly has been presented. It is different from the chronic decline since it relates to epilepsy that has a late onset, where the decline is more abrupt and occurs soon after the diagnosis of epilepsy. Also, it does not seem to degrade the crystalized cognitive functions such as long-term memory and language abilities; but rather the fluid intelligence: perceptual reasoning and processing speed. This fast mental ageing is coined 'accelerated cognitive ageing', or ACA [7, 8]. Even though the clinical characteristics of the ACA comorbidity has been described, the brain mechanism of action of the pathology remains unclear. This is why in this study we assess, using fMRI, different dynamic parameters of large-scale networks that could help describing neuronal mechanisms behind the decline.

The brain has been proven to efficiently work in functionally connected networks, i.e. the aggregation of multiple region of interests (ROIs) [9, 10]. Also, the links between brain activity and behavior/cognition are often better analyzed through the use of large-scale networks [11–14]. Therefore, the resting-state network (RSN) dynamics are informative features in neuro-psychiatric disorders and can be used as neurobiomarkers. Between-network neurodynamics analysis can be performed using network time series from a spatial independent component analysis (ICA), a reliable data-driven method [15, 16]. In a seed-based approach (voxel-to-voxel, or ROI-to-ROI), the seeds or ROIs have to be defined a priori, and this can be quite inter-subject variable, especially for resting-state (task-free) fMRI. Hence the choice of ICA decomposition for extracting the functional brain RSNs and their activity time courses is favorable for robust rs-fMRI group analysis, and is preferred here.

After extracting the time series per network, diverse methods can be applied to estimate the dynamics of these networks. Currently, to find the abnormalities in brain functioning of patients with neuropsychiatric disorders, research has mainly focused on static functional connectivity (sFC), i.e. pairwise correlations between the functional networks' activity. It has been shown that at rest the default-mode network is involved in the impairment in autism [17, 18]; and the attention and executive networks in attention deficit hyperactivity disorder (ADHD) [19, 20]. But for more complex mental illnesses (e.g. depression or schizophrenia) dynamic measures of the brain connectivity seem to be more robust biomarkers [21, 22].

In that regard, effective connectivity (EC) can be used [23, 24]. The EC, or measure of causality in brain connectivity, has already shown promising results for autism and depression [25, 26]. However, in terms of cognitive decline in the ageing process, clear causal dynamics indicators are missing. Liu et al. have shown links between cognitive decline and changes in Granger causality (a measure of EC) of cognitive networks in mild cognitive impairment [27]. Using a different—more hypothesis driven—causal measurement, the dynamic causal

modeling (DCM), Tsvetanov et al. showed that differences in between and within large-scale resting-state networks were linked to cognitive performance in elderly [28]. More precisely they demonstrated that the DCM of the default mode, dorsal attention and salience networks (DMN, DAN, SN) where associates with increased influence on cognitive functioning—that tends to decline—in healthy elderly. Another study showed the involvement of DMN and SN in the cognitive decline in cognitively impaired elderly using also the DCM approach [29].

More recently, to assess neuro- and psycho-pathology, the dynamic functional connectivity (dFC) methods have been used as well [21, 30–34]. These are similar to the sFC but when correlation coefficients are calculated over different time windows. By sliding these windows throughout the full time series, one can extract the dFC between the networks being assessed [35, 36]. This method of analysis has been for example applied to extract abnormal brain states in the study of schizophrenia, which has stronger classification power as compared to classic static functional connectivity [37]. The same methods can also be applied in the frequency domain, and the sliding windows approach can be applied on Fourier spectra, giving rise to spectrograms. And similar spectro-temporal analysis of wavelet coherence has not been much used in fMRI studies. Chang and Glover have shown added value in assessing non-stationarity and time-varying anti-correlation between DMN and cognitive networks, using wavelet coherence [38]. To relate to cognitive decline, only one study used wavelet coherence analysis of near-infrared spectroscopic signals to infer changes in brain functional connectivity in respect to performance in vigilance task, suggesting that decreased attention—a known phenomenon in ageing, and probably associated to cognitive decline—was attributed to reduced phase coherence between left prefrontal region and sensorimotor area. In a previous rs-fMRI study, we have successfully shown that coherent patterns, using wavelet coherence, are also impaired in autism; and the duration of these coherent patterns can be used as biomarker of the disorder [39].

Finally, evaluation of brain dynamics can be done through an evolutionary game theory approach on brain networks (EGN). Madeo and his colleagues were first to implement the methods on BOLD fMRI signals [40]. They demonstrated that EGN model is able to mimic functional connectivity dynamics, which could then be used to simulate the impact of brain network lesions onto the networks' dynamics. This was done using simulated time series and real fMRI data from healthy subjects. Here, we present the first fMRI study where the EGN model is applied to a clinical population in order to see if network emulative powers (parameters of the model) can be descriptors of ACA.

The goal of our research is to define the most informative fMRI neurobiomarkers for ACA based on neurodynamics metrics. To achieve this, we use these metrics as features and train classifiers for ACA detection. The features that lead to the best classification results can then be considered the best descriptors of ACA. Therefore, in our study, the aforementioned neurodynamics techniques are used to extract fMRI-base dynamic features of brain networks, which are then fed to classifiers, to train and validate them. Decision tree, linear discriminant analysis, the K-nearest neighbors and the support vector machine are common supervised machine learning algorithms that divide the training data in two or more categories. Counting the number of correct binary classifications, in the validation dataset, leads to a certain amount of accuracy, i.e. the percentage of participants that has been correctly assigned to their respective label (in our case, ACA or non-ACA/controls); which shows the descriptive power of the features, and to a certain extent its usability in a clinical environment. Convolutional neural network (CNN) can also be used to extract and test specific brain features using resting-state fMRI. It has already been tested in order to help diagnosing neuropsychiatric disorders, such as autism [41], ADHD [42], and schizophrenia [43]. An important drawback of this classification technique is that it usually needs a large amount of data to be trained upon; Moreover, the

understanding and explanation of the feature maps obtained at different CNN layers can be arduous and troublesome. This is why we used here the aforementioned supervised machine learning algorithms and assessed the performance of classifiers using neurodynamics metrics, in order to find the best biomarkers of the ACA comorbidity, which if replicated could help describing the mechanism of action of the cognitive decline. Testing the classification pipeline we describe here, in a bigger dataset stratified, e.g. through duration since onset of epilepsy or age ranges, could even help predict the course of the decline and its severity.

## 2 Materials and methods

Fig 1 provides an overview of the Materials and methods section, and short descriptions of the steps of the study pipeline. The study was approved by the Dutch Medical Research Ethical Committee (MREC) of Maxima Medical Center. All subjects signed informed consent and gave permission to use the clinical data for scientific purposes.

### 2.1 Dataset, preprocessing, and brain network extraction

The dataset contains two resting-state, RS1 and RS2, sessions acquired from 10 ACA patients and 14 controls. Prior to RS2, a silent word generation task was performed for 7 min. Dynamics features that show significant differences between the two populations in RS1 as well as in RS2, would show robustness of the features as biomarkers. Indeed, since the task in-between resting-states was cognitively demanding, such features would have discriminant power regardless of the brain fatigue the participants experienced. On the contrary, the difference in findings between RS1 and RS2 could shed light on the cognitive reserve capacity of the ACA patients, which can be of interest too. Table 1 shows the demographics of the participants, the type of epilepsy, and the deterioration scores, i.e. cognitive decline, for Full-Scale IQ, Perceptual Reasoning Index, and Verbal Comprehension Index (respectively FSIQ, PRI, and VCI). Deterioration scores were calculated by subtracting the estimated premorbid IQ scores from the actual scores, i.e. WAIS-IV (actual) IQ-scores—OPIE-IV (premorbid) IQ-scores [44, 45].

The T1-weighted anatomical images were acquired using a 3.0 T imaging system (Philips Achieva) with a 3D Fast Field Echo (FFE) sequence: echotime (TE) = 3.8 ms, repetition time (TR) = 8.3 ms; Inversion time (TI) = 1035 ms; field of view (FOV) = 240x240 mm$^2$, with 180 sagittal slices; flip-angle (FA) = 8 deg.; and voxel size = 1x1x1 mm$^3$; with SENSE factor of 1.5, and scan duration of 6:02 min. Functional MRI-data were acquired using multi-echo echo-planar imaging (ME-EPI) sequences with 3 echoes: TEs = 12, 35, 58 ms, TR = 2 s; FA = 90 deg.; SENSE factor = 2.7; 208 dynamics for a total of 7 minutes; 27 axial slices (with no gap), 64x64 matrix FOV, with a 3.5x3.5x4.5 mm$^3$ voxel size. The raw multi-echo fMRI data were first

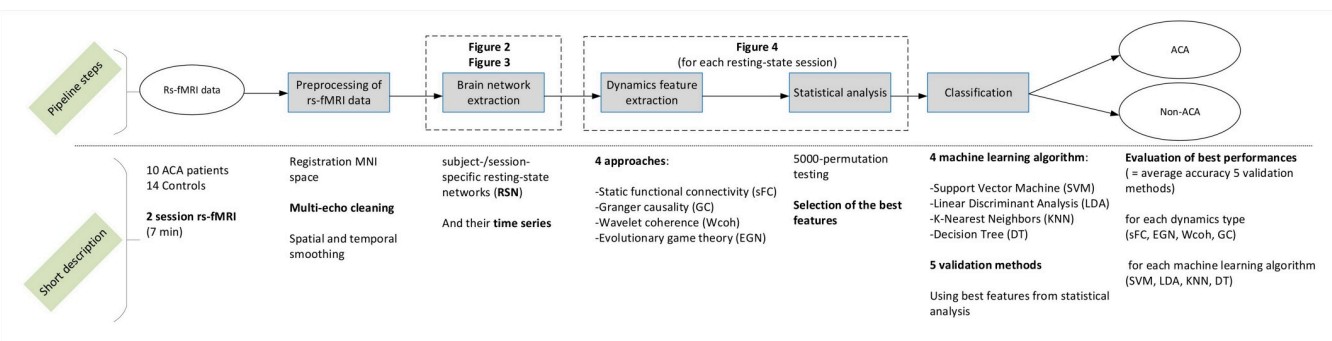

**Fig 1. Overview of the methodology.**

**Table 1. Demographic and clinical information of the participants.**

|  | ACA | Healthy controls |
|---|---|---|
| **Age in years M (SD) range** | 61.3 (8.9) | 62.2 (9.8) |
|  | 50–74 y | 47–79 y |
| **Gender** | 70.0% male | 35.7% male |
| **Handedness** | 100.0% right-handed | 92.9% right-handed |
| **Age at epilepsy onset M (SD) + range** | 35.0 (14.6) | - |
|  | 15–59 y |  |
| **Duration of epilepsy M (SD) + range** | 22.3 (15.2) | - |
|  | 1–50 y |  |
| **Type of epilepsy** | 40.0% cryptogenic localization-related | - |
|  | 40.0% symptomatic |  |
|  | 20.0% idiopathic |  |
| **Dominant seizure type [a]** | 10.0% simple partial | - |
|  | 20.0% complex partial |  |
|  | 20.0% absence |  |
|  | 0.0% tonic-clonic |  |
|  | 50.0% seizure free |  |
| **Status epilepticus** | 50.0% yes |  |
| **Seizure frequency** | 50.0% seizure (sz) free | - |
|  | 0.0% < 1 sz/y |  |
|  | 20.0% 1–5 sz/y |  |
|  | 20.0% 1 sz per 2 months |  |
|  | 0.0% monthly sz |  |
|  | 10.0% weekly sz |  |
|  | 0.0% daily sz |  |
| **Drug load [b]** | 1.5 (0.4) | - |
| **WAIS-IV indexes** |  |  |
| FSIQ | 76.7(8.7)* | 108.4(13.4) |
| VCI | 94.7 (10.6) | 106.2 (12.2) |
| PRI | 75.8 (6.9)* | 103.2 (13.8) |
| WMI | 79.6 (10.0)* | 104.9 (13.3) |
| PSI | 67.0 (14.4)* | 114.6 (8.7) |
| **Deterioration scores[c]** |  |  |
| Det-FSIQ | -22.1 (5.0)* | -1.3 (8.2) |
| Det-VCI | -0.5 (8.9) | 0.2 (6.0) |
| Det-PRI | -21.0 (4.5)* | 0.0 (8.7) |
| **Memory scores** |  |  |
| Auditory | 93.9 (9.7)* | 109.9 (12.6) |
| Visual | 92.7 (7.0) | 102.8 (10.9) |
| Delayed memory | 93.5 (8.4)* | 106.3 (10.9) |

Note: * = $p < 0.01$ sign. difference between groups. WMI: Working Memory Index. PSI: Processing Speed Index.

[a] Dominant seizure type is determined for the two years preceding neuropsychological assessment.

[b] The prescribed daily dose of antiepileptic medication divided by the defined daily dose.

[c] The deterioration scores = [WAIS-IV (actual) IQ-scores—OPIE-IV (premorbid) IQ-scores]

preprocessed following the pipeline of [46], using the python script *meica.py* (available at www.bitbucket.org/prantikk/me-ica). Multi-echo- (ME)-ICA cleaning was applied in order to denoise each individual fMRI scan [47]. ME-ICA cleaning has been proven to be the most

robust denoising method for resting-state fMRI, and tend to improve substantially effect sizes and statistical power [48, 49]. Multi-echo cleaned data were further processed using FEAT (FMRI Expert Analysis Tool) Version 6.00, part of FSL (FMRIB's Software Library, www.fmrib.ox.ac.uk/fsl). The following pre-statistics processing was applied; spatial smoothing using a Gaussian kernel of FWHM 5.0mm; grand-mean intensity normalization of the entire 4D dataset by a single multiplicative factor; high-pass temporal filtering (Gaussian-weighted least-squares straight line fitting, with sigma = 50.0 s). The ME-ICA cleaning preprocessing led to an average, among all participant/scans, of 18 activity-related independent components (ICs or networks). Therefore, we chose 18 degrees of freedom for our group-ICA decomposition. The 18 spatial maps corresponding to functional resting-state networks (RSNs) were extracted using group-ICA followed by dual regression as implemented in FSL [50, 51]. After the dual-regression steps, we obtained all subject/session-specific RSN maps and their associated time series. This brain network extraction step is depicted in Fig 2.

After visual inspection and a goodness-of-fit function implementation [52], to match these maps with the RSN template from [11], 4 noise-related RSNs were discarded, and 14 relevant functional brain networks were kept for further analyses. The time series associated with these 14 RSNs were then used to extract our brain networks features as explained in the next section. The 14 networks, or Independent Components (ICs), and their labelling are depicted in Fig 3.

## 2.2 Static connectivity, causality, phase-coherence and emulative power

First, static functional connectivity (sFC) analysis was performed on all pairs of brain network signals, using Pearson correlation. This is applied for each participant's RSN time series, and each session (RS1 and RS2), leading to two times 24 (14 controls; 10 patients) correlation matrices. The total amount of sFC features per subject/session (unique entries of sFC matrix) was 91 (= 14x13/2; the connectivity matrix being symmetrical, and its diagonal discarded).

Second, using the same ICs (or RSNs) time series we calculated the Granger causality (GC) matrices for each individual and resting-state session using the Matlab toolbox of [53]. A GC connectivity matrix is defined as $\boldsymbol{G} = \{g_{ij}\} \in \mathbb{R}^{N \times N}$ ($N$ = numbers of networks), where each matrix entry $g_{ij}$ represent the directed causal connectivity from network $j$ towards network $i$. More specifically $g_{ij} = F_{Y \to X|Z}$, which is the GC between signal Y (time series of network $j$) and X (time series of network $i$) conditional on Z (remaining time series of networks $k \neq i \neq j$)— more details in [54]. The upper triangle of $\boldsymbol{G}$, i.e. $\{g_{ij}\}, \forall j > i$ represent the inward causal pairwise connectivity $g_{ij}$, while the lower triangle, i.e. $\{g_{ji}\}, \forall i > j$ is all the outward causal connections of $\{g_{ij}\}$. Since GC matrices are asymmetrical (directed connectivity) by subtracting those

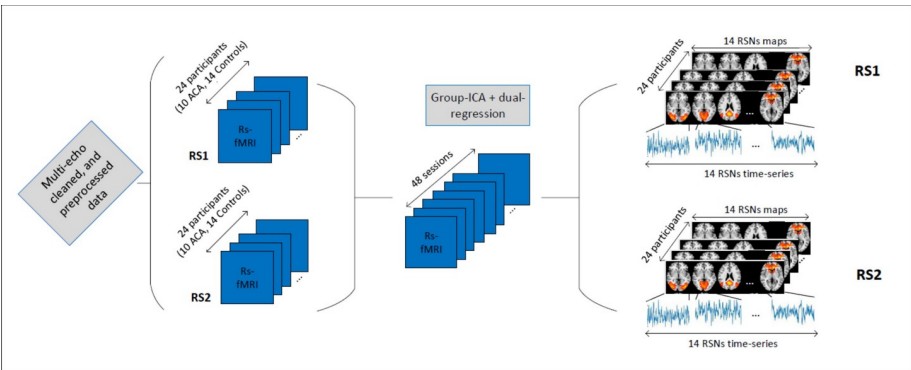

**Fig 2. Brain networks (RSN) time series extraction.**

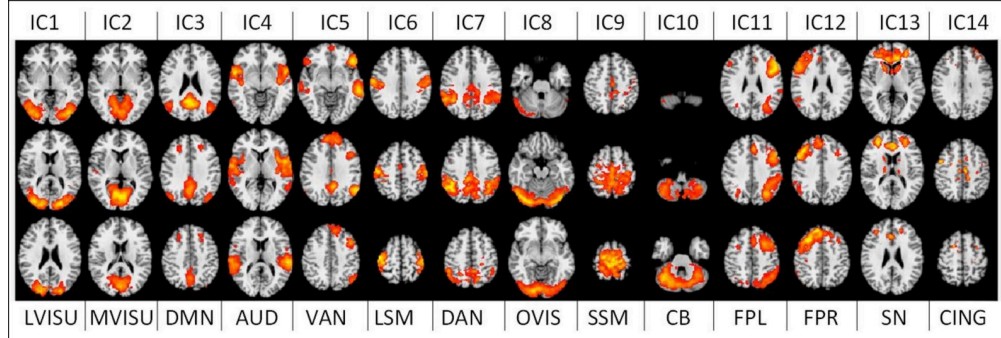

**Fig 3. 14 large-scale brain functional resting-state networks (RSNs) used in the study.** LVISU–lateral visual network; MVISU–medial visual network; DMN–default mode network; AUD–auditory network; VAN–ventral attention network; LSM—lateral sensory-motor network; DAN–dorsal attention network; OVIS–occipital visual network; SSM–superior sensory-motor network; CB–cerebellar network; FPL–left fronto-parietal network; FPR–right fronto-parietal network; SN–salience network; CING–the cingulate network.

triangular matrices, i.e lower-upper, we can extract the non-null net pairwise GC triangular matrix **NetGC** = {$netg_{ij}$} = {$g_{ij}$—$g_{ji}$}, $\forall\, i > j$. We also extracted the In- and Out-causal degree of each network by summing respectively the columns and rows of the **G** matrices, i.e. **DegIn** = {$\sum_{j=1}^{N} g_{ij}$} $\in \mathbb{R}^{N}$, and **DegOut** = {$\sum_{j=1}^{N} g_{ij}$} $\in \mathbb{R}^{N}$. We finally derived the net (out-in) causal degree of each network, i.e. **NetDeg = DegOut—DegIn$^T$**. Since we have $N$ = 14 networks, the total amount of GC features was 315 (= 3x91 (in-, out-, net-GC triangular matrices) + 3x14 (DegIn, DegOut, and NetDeg vectors)). These GC feature matrices and vectors were calculated per subject and session, and are illustrated in Fig 4B.

Thirdly, we applied the wavelet-coherence approach with the following processing steps: (i) extraction of spectro-temporal maps (also called scalograms) of significant phase coherence, i.e. localized (in time and frequency/scales) significant phase-locked correlation between pairs of signals, as in [55]. The phasing between the signals is expressed in radians and extracted from $\theta = arg(Wxy)$ where $Wxy$ is a complex number defining the cross-wavelet transform between the two network signals being assessed. The phase information are summarized in 4 categories: in-phase when $\theta \epsilon$ ($-\pi/4$, $+\pi/4$); anti-phase when $\theta \epsilon$ ($3\pi/4$, $-3\pi/4$); and signal 1 leading (or lagging) signal 2 when $\theta \epsilon$ ($\pi/4$, $3\pi/4$) (or $\theta \epsilon$ ($-3\pi/4$, $-\pi/4$)); for simplification we colored the phase information in the wavelet-coherence maps (see Fig 4C). For each of the 5 period scales ([4, 8) s; [8, 16) s; [16, 32) s; [32, 64) s; and [64, 128] s) we extracted the time of coherence (in % of the scan duration) between the 91 pairs of networks. More details on the time of coherence metric in [39]. The time of coherence extraction was performed for all of the 4 aforementioned phases. Therefore, in total we obtained 1820 (= 4x5x91) coherence-based features per subject's session (Fig 4C).

Finally, the Evolutionary Game theory on Network (EGN) was implemented as in [40]. Instead of trying to predict brain signal (time series) dynamics and correlating them to the original BOLD time series, we here simply took the EGN-connectivity matrix **A** = {$a_{ij}$} $\in \mathbb{R}^{N \times N}$ ($N$ = numbers of networks), and derived the vectors of emulative powers (EPs). Briefly, each entry of **A**, $a_{ij}$, described the pairwise and directed ($i{\rightarrow}j$) weight of the network $i$ in its ability to emulate (positive value, $a_{i,j} > 0$) or not (negative value, $a_{ij} < 0$) the activity of the network $j$. **A** can be seen as similar to the GC connectivity matrices, albeit not in the sense of causality but in terms of replication mechanism. From these matrices of similar size ($N^2$) as the sFC and GC matrices, we calculated our novel metrics, namely the In-EPs, Out-EPs, and Net-EPs, per participants and session (Fig 4D). The Out-EP of each network, i.e. its power to emulate

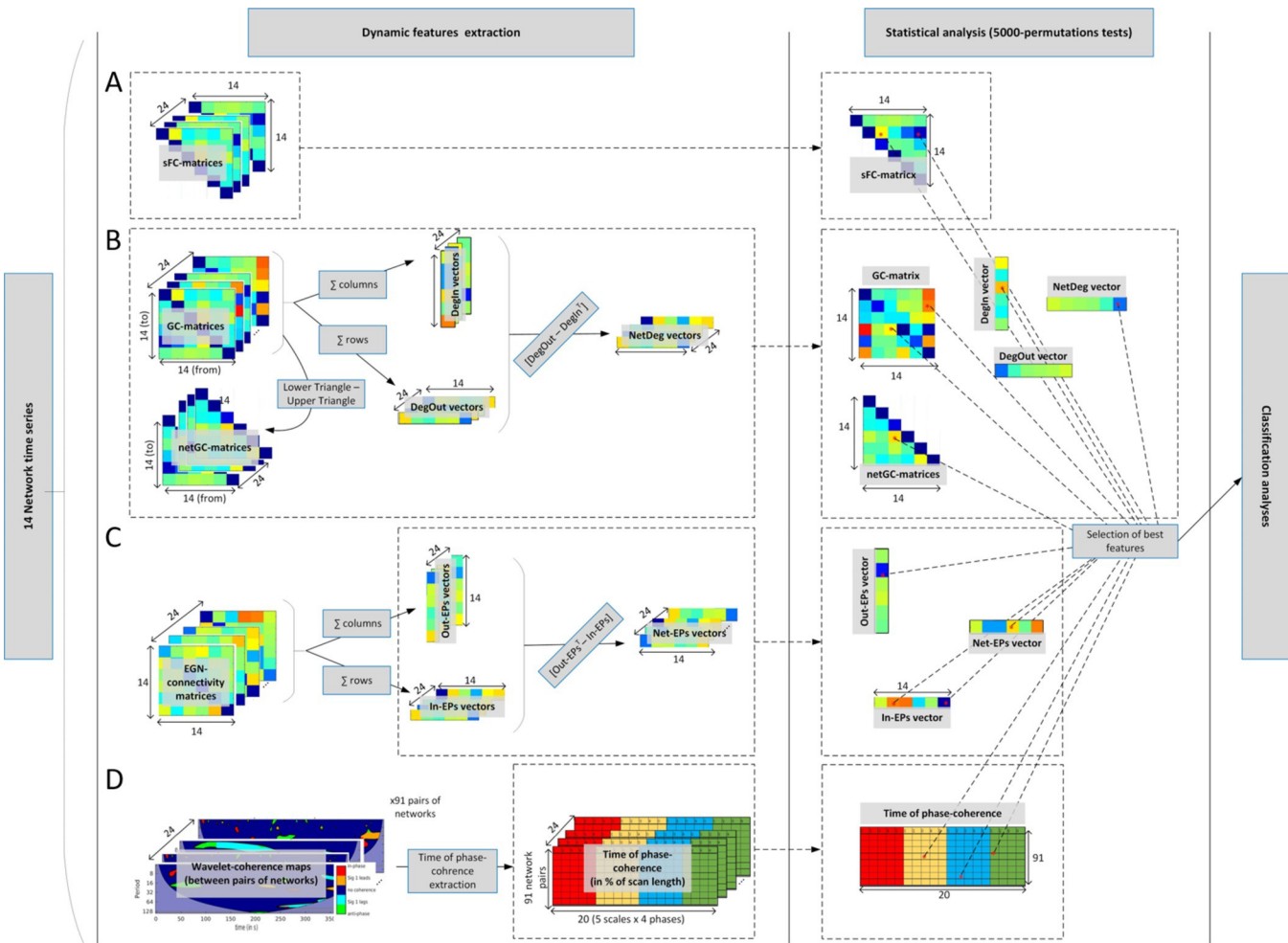

**Fig 4. Network feature extraction and statistical analysis.** A: extraction of the static functional connectivity features. B: computation of the Granger causality matrices and vectors. C: Emulative power vectors derived from the EGN method; D: Wavelet coherence approach and the time-of-coherence metric extraction.

(replicate) the other networks' activity is calculated by summing the columns of the EGN-connectivity matrix, i.e. $OutEP_i = \sum_{j=1}^{N} a_{ij}$. The power of one network to be emulated by the other networks is represented by its in-EP, that is, the sum of the EGN-connectivity matrix rows, i.e. $InEP_j = \sum_{i=1}^{N} a_{ij}$. And the net replicative power of a network is defined by the Net-EP, i.e. **$NetEP = OutEP^T — InEP$**. Since $N = 14$, we ended up with three vectors of 14 EPs per person's rs-fMRI session, i.e. 42 features (Fig 4D). The network feature extractions were repeated for the two resting-state sessions.

## 2.3 Selection of the most relevant features, and classification analysis

After obtaining the subject-/session-specific sFC, GC, time of phase-coherence matrices, and EP vectors, we performed non-parametric permutations tests (5000-permutations) for each entry of the mentioned matrices/vectors. The resulting matrices and vectors were then thresholded at 5% level ($p < 0.05$), giving us the list of pairs of networks where the sFC, GC, time of coherence, and EPs were significantly different between the two groups (ACA and Controls). This statistical analysis was performed to reduce the total number of fMRI-based features to be

used by the classifiers, which was initially 2268 (91 sFC, 315 GC, 1820 WCoh, 42 EGN). This statistical analysis to obtain the best features is depicted in Fig 4.

In order to assess the strength of the abovementioned neurodynamics methods in describing an ACA in patients with epilepsy, classifiers were trained and validated. We used 4 types of machine learning, namely, the support vector machine (SVM), the linear discriminant analysis (LDA), the k-nearest neighborhood (KNN), and the decision tree (DT) algorithms. To validate the classifiers, 5 methods were used: validation on RS2 features (with classifiers trained on RS1's); validation on RS1 features (with classifiers trained on RS2's); leave-on-out cross-validation (LOOCV) in RS1 (24-folds cross-validation); LOOCV in RS2 (24-folds); LOOCV in RS1 and RS2 concatenated (48-folds). 5 validation methods are used to allow us to fully trust the performances of the classifiers. It is also more robust: we can assess consistency and avoid depicting unreproducible 'chance' performances. For each of the classifier types of training and validation, different set-ups of features were used. First, we tested the classifiers using the best features (after statistical analyses) from one type of dynamics at a time, i.e. only best features from sFC analysis, then only significant features from GC analysis, and so on. Next, we refined these set-ups to the use of features that fell into two categories: (i) features/predictors that were used in the DT algorithm, i.e., selected by the Classification And Regression Trees (CART; default split predictor selection technique for classification trees in Matlab); (ii) features that involved the DMN, DAN or SN, since these networks are often mentioned in ageing and cognitive decline literature Then we reiterated refinements of the classifier setups by removing or adding features that still comply with the aforementioned criteria (i and ii), if they were improving classification accuracy in the previous refinement iterations. Such a search for the best combinations of features led us to a dozen of classification analyses per dynamics metrics. Finally, we tested classifiers with combined network features, e.g. using the best GC-based features combined with the best WCoh-based features. We refined the classification models the same way it was performed in the classification analyses per network connectivity metric. We also tried the combination of all best features, i.e. combining all features selected by the statistical analyses of all static and dynamics metrics. In the combined network features setups, a dozen (through refinements) classifiers were also tested. For each of the classifier setup tested (~60 = 5x12), we reported the accuracy, sensitivity and specificity. For clarity, in the Results section, only the best classifiers performances per feature type, and per machine learning algorithms are reported: by looking at the highest average accuracy of the classifiers over the 5 validation methods—mentioned at the beginning of this paragraph.

## 3 Results

### 3.1 Selection of the best network features

After extracting all static and dynamic features (sFC, GC, WCoh, EGN), 5000-permutation tests between the two groups' features were applied and thresholded at (uncorrected) p<0.05, giving us the most significant differences between the groups, i.e. the 'best' features. The results of the significant permutation tests for sFC-, GC- and EGN-based features are shown in Supporting Information (S1-S4 Figs in S1 File).

Regarding the WCoh-based features, namely the time-of-coherence measurements, the permutation tests gave us an average of more than 50 significant features per phase (in-phase, anti-phase, leading, and lagging). In order to avoid an overfitting problem, we reduced the number of best features even further, as follows: (i) discarding features that also showed significant differences in controls when comparing RS1 and RS2; (ii) keeping features that showed robustness, i.e. significant in several scales, or in multiple phases; or (iii) time-of-coherence with higher evidence of between-group differences (p<0.01). This led us to having an average

of 10 best features per phase. An example of a pairwise in-phase time-of-coherence that complied with the aforementioned criteria is depicted in Fig 5.

All of the best features for each neurodynamics metric and their labels are summarized in Table 2.

## 3.2 Classification performances

After static (sFC) and dynamic analyses (GC, WCoh, EGN) followed by the statistical permutation testing we obtained the best descriptors (features) for each metric. All and subsets of these best features were used for training and validating classifiers. In Table 3, the best performances of each individual feature type are shown. For wavelet-coherence, the in-phase time of coherence were the best descriptors of ACA, reaching an average 77.1% accuracy with the decision-tree algorithm, using DMN, SN and visual networks. Using 4 DMN-based and 2 SN-based static connectivity features, the average classification accuracy was 77.9%, with LDA algorithm. With the emulative powers (EGN-based dynamics metric) of the DMN and the DAN we could reach 82.9% accuracy, also using LDA algorithm. Note that the Out-EP and Net-EP of the DAN were not part of selected best features (Table 2), but interestingly improved the classification accuracy when added. This phenomenon can be explained by the fact that, even though Out- and Net-EPs of the DAN are not statistically significant (p>0.05), the way they are pooled with their significant counterparts (In-EPs) in SVM and LDA algorithms, can allow classifiers to be more discriminative. GC-based features, however, led only to 67.9% accuracy with DT algorithm. Overall the metrics giving the best results for classification can be ranked as EPs > sFC > in-phase coherence > GC.

After assessing the best features of each metric, we also trained and tested classifiers with combinations of the features. This gave us the best performances for each classifier type. LDA reached 85% average accuracy using all 86 best features (Table 2). KNN led to an average of 80.8% accuracy with a mix of 82 features that is the 86 best features minus 4 EPs: Out-EPs of DMN and OVIS; and Net-EPs of DMN and CING. DT algorithm reached 80.0% with only 6 features (3 Coherence, 2 EGN, and 1 sFC). And SVM had the best results with 15 features (2 in-phase coherences, 4 in-EPs, 3 GC, and 6 sFC) reaching average accuracy of 90.0%. In the latter SVM set-up, it is worth noting that the independent training and validation, i.e. RS1-trained RS2-validated, and vice-versa, could classify ACA at 91.7% accuracy; the 48-fold LOOCV had 95.8% accuracy. Finally, the overall best average accuracy (81.0%), over all the

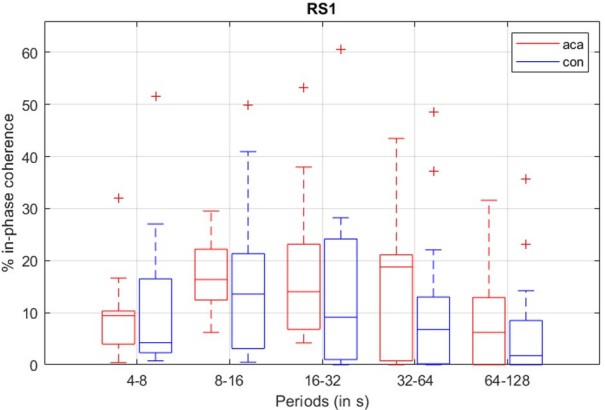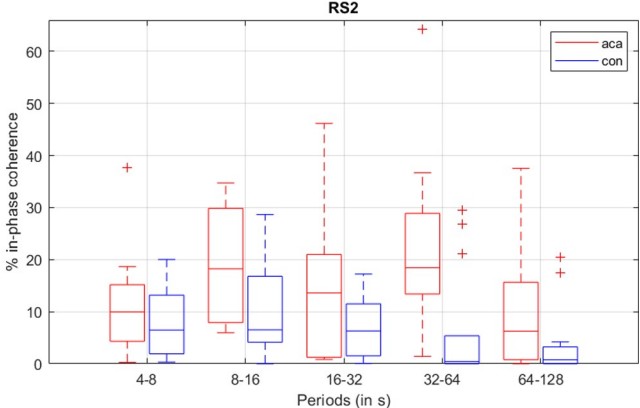

**Fig 5. Boxplots of the time of in-phase coherence (in % of scan duration = 7min) for the network pair MVISU-DMN.** red boxplot–ACA; blue boxplot–controls; left–RS1; right–RS2. Middle line in the box–median value; box size– 25$^{th}$ to 75$^{th}$ percentiles; whisker limits–most extreme values that are not outliers; '+'–outliers.

**Table 2. 86 best features used in the classification analyses.**

| Wavelet-coherence | | Granger-causality | | static FC | EGN |
|---|---|---|---|---|---|
| *in-phase time-of-coherence* | *leading time-of-coherence* | *net1→net2 GC* | *net1←net2 GC* | *net1↔net2* | *EPs* |
| ip_LVISU-OVIS_s1 | lead_AUDI-FPR_s1 | MVISU->LVISU | MVISU<-LSM | sFC-DMN-VAN | In-EP_DMN |
| ip_LSM-CING_s1 | lead_MVISU-DMN_s1 | MVISU->FPL | MVISU<-FPR | sFC-MVISU-DMN | In-EP_DAN |
| ip_OVIS-SN_s2 | lead_MVISU-OVIS_s2 | MVISU->CING | LVISU<-AUDI | sFC-DMN-LSM | In-EP_SSM |
| ip_CB-SN_s2 | lead_DMN-OVIS_s2 | DMN->AUDI | LVISU<-SSM | sFC-DMN-CB | In-EP_CING |
| ip_MVISU-DMN_s2 | lead_LSM-OVIS_s2 | DMN->SN | LVISU<-SN | sFC-AUDI-CING | Out-EP_DMN |
| ip_OVIS-SN_s3 | lead_LSM-OVIS_s3 | VAN->CING | VAN<-DAN | sFC-LSM-OVISU | Out-EP_OVIS |
| ip_CB-SN_s3 | lead_OVIS-CB_s3 | DAN->FPR | VAN<-AUDI | sFC-DAN-SN | Net-EP_DMN |
| ip_DMN-DAN_s4 | lead_LVISU-VAN_s4 | AUDI->SN | DAN<-CING | sFC-OVISU-SN | Net-EP_CING |
| ip_MVISU-DMN_s4 | lead_DMN-VAN_s4 | LSM->CING | AUDI<-SSM | sFC-CB-FPL | |
| *anti-phase time-of-coherence* | lead_FPL-SN_s4 | FPL->FPR | *net pairwise GC* | sFC-CB-CING | |
| ap_AUDI-FPR_s1 | *lagging time-of-coherence* | FPL->CING | netgc_MVISU->LVISU | | |
| ap_VAN-FPL_s1 | lag_LVISU-DAN_s1 | *network DegIn, DegOut,* | netgc_MVISU->CING | | |
| ap_AUDI-SSM_s1 | lag_OVIS-SSM_s1 | *and netDeg* | netgc_LVISU->LSM | | |
| ap_DMN-VAN_s2 | lag_OVIS-SN_s2 | GC_DegOut_LSM | netgc_LVISU->SN | | |
| ap_OVIS-SN_s2 | lag_OVIS-FPL_s2 | GC_NetDeg_DAN | netgc_DMN->VAN | | |
| ap_LVISU-VAN_s3 | lag_CB-FPR_s2 | GC_NetDeg_AUDI | netgc_DMN->AUDI | | |
| ap_OVIS-SN_s3 | lag_CB-FPL_s2 | GC_NetDeg_LSM | netgc_DAN->CING | | |
| ap_VAN-SN_s4 | lag_LSM-OVIS_s3 | | netgc_FPL->FPR | | |
| ap_DMN-AUDI_s5 | lag_VAN-CB_s3 | | | | |

ip–in-phase; ap–anti-phase; lead–leading; lag–lagging; s1 –scale 1 = [4, 8] s; s2 –scale 2 = [8, 16] s; s3 –scale 3 = [16, 32] s; s4 –scale 4 = [32, 64] s; s5 –scale 5 = [64, 128] s; →(←)–direction of GC; netgc–net pairwise GC (out-in); GC_DegOut–GC out degree of a network; GC_DegIn–GC in degree of a network; sFC–pairwise static function connectivity; ↔ –bidirectional connectivity for sFC; EGN–evolutionary game theory on network; EPs–emulative powers; In-, Out-, Net-EP–in, out, net emulative power of a network; net1(2)–network 1 (or 2).

machine-learning algorithms, was obtained using even fewer features (10: 2 in-phase coherence, 2 in-EPs, and 6 sFC). These results are depicted in Table 4. Looking at the classifier algorithms, in general, SVM was performing the best. With few features (< 10), DT was performing quite well, but still not as good as SVM and LDA. KNN was overall the least accurate classification algorithm. It is worth noting that in some set-ups DT algorithm reached 100% accuracy, such as for the LOOCV of RS2 with the mix of 6 features (Table 4); but this was not consistent, since in the same set-up, the RS1-trained-and-RS2-validated classifier reached only 66.7% accuracy.

Overall, EGN-based net emulative powers are the best descriptors, especially when using DMN and DAN features. And when those are combined with specific time of in-phase coherence and static connectivity (correlation) values, SVM algorithm outperforms the other algorithm with an average of 90.0% accuracy; with peaks at 91.7% for the RS-independent training/validation classifications, and 95.8% in the 48-folds LOOCV. For a more intuitive visualization of the results presented above, we graphically summarize Tables 3 and 4 in Fig 6A and 6B respectively.

## 4 Discussion

### 4.1 Clinical relevance

We have shown that the emulative powers (Eps) of the default mode network and the dorsal attention networks are biomarkers of ACA. Combining these features with DMN and SN-

**Table 3. Summary tables of the best classifier performances for each network connectivity measures: EGN, coherence, sFC, and GC.**

| | List of feature used | | Validation methods | SVM | | | LDA | | | KNN | | | DT | | |
|---|---|---|---|---|---|---|---|---|---|---|---|---|---|---|---|
| | | | | acc | sens | spec | acc | sens | spec | acc | sens | spec | acc | sens | spec |
| EGN | In-EP_DMN | | tRS1_vRS2 | 0.917 | 0.900 | 0.929 | 0.917 | 0.900 | 0.929 | 0.708 | 0.500 | 0.857 | 0.750 | 0.700 | 0.786 |
| | In-EP_DAN | | tRS2_vRS1 | 0.708 | 0.700 | 0.714 | 0.833 | 0.800 | 0.857 | 0.708 | 0.600 | 0.786 | 0.583 | 0.600 | 0.571 |
| | Out-EP_DMN | | Loocv_RS1 | 0.750 | 0.700 | 0.786 | 0.750 | 0.700 | 0.786 | 0.375 | 0.300 | 0.429 | 0.583 | 0.600 | 0.571 |
| | Out-EP_DAN | | Loocv_RS2 | 0.750 | 0.700 | 0.786 | 0.833 | 0.800 | 0.857 | 0.625 | 0.500 | 0.714 | 0.542 | 0.300 | 0.714 |
| | Net-EP_DMN | | Loocv_conca_RS | 0.771 | 0.750 | 0.786 | 0.813 | 0.750 | 0.857 | 0.708 | 0.600 | 0.786 | 0.771 | 0.700 | 0.821 |
| | Net-EP_DAN | | **Average** | 0.779 | 0.750 | 0.800 | 0.829 | 0.790 | 0.857 | 0.625 | 0.500 | 0.714 | 0.646 | 0.580 | 0.693 |
| | | | | acc | sens | spec | acc | sens | spec | acc | sens | spec | acc | sens | spec |
| WCoh | | | tRS1_vRS2 | 0.417 | 1.000 | 0.000 | 0.917 | 0.800 | 1.000 | 0.583 | 0.700 | 0.500 | 0.750 | 1.000 | 0.571 |
| | ip_OVIS-SN_s2 | | tRS2_vRS1 | 0.417 | 1.000 | 0.000 | 0.708 | 0.700 | 0.714 | 0.750 | 0.900 | 0.643 | 0.667 | 0.500 | 0.786 |
| | ip_MVISU-DMN_s2 | | Loocv_RS1 | 0.417 | 1.000 | 0.000 | 0.417 | 1.000 | 0.000 | 0.625 | 0.600 | 0.643 | 0.708 | 0.800 | 0.643 |
| | | | Loocv_RS2 | 0.417 | 1.000 | 0.000 | 0.792 | 0.700 | 0.857 | 0.875 | 0.900 | 0.857 | 1.000 | 1.000 | 1.000 |
| | | | Loocv_conca_RS | 0.417 | 1.000 | 0.000 | 0.729 | 0.650 | 0.786 | 0.729 | 0.750 | 0.714 | 0.729 | 0.750 | 0.714 |
| | | | **Average** | 0.417 | 1.000 | 0.000 | 0.713 | 0.770 | 0.671 | 0.713 | 0.770 | 0.671 | 0.771 | 0.810 | 0.743 |
| | | | | acc | sens | spec | acc | sens | spec | acc | sens | spec | acc | sens | spec |
| sFC | sFC-DMN-VAN | sFC-DMN-CB | tRS1_vRS2 | 0.875 | 0.700 | 1.000 | 0.875 | 0.700 | 1.000 | 0.875 | 0.800 | 0.929 | 0.458 | 0.400 | 0.500 |
| | sFC-MVISU-DMN | sFC-DAN-SN | tRS2_vRS1 | 0.750 | 0.900 | 0.643 | 0.792 | 0.900 | 0.714 | 0.750 | 1.000 | 0.571 | 0.542 | 0.700 | 0.429 |
| | sFC-DMN-LSM | sFC-OVISU-SN | Loocv_RS1 | 0.667 | 0.600 | 0.714 | 0.667 | 0.600 | 0.714 | 0.583 | 0.400 | 0.714 | 0.625 | 0.500 | 0.714 |
| | | | Loocv_RS2 | 0.833 | 0.700 | 0.929 | 0.750 | 0.600 | 0.857 | 0.750 | 0.500 | 0.929 | 0.833 | 0.700 | 0.929 |
| | | | Loocv_conca_RS | 0.771 | 0.700 | 0.821 | 0.750 | 0.700 | 0.786 | 0.750 | 0.550 | 0.893 | 0.583 | 0.500 | 0.643 |
| | | | **Average** | 0.779 | 0.720 | 0.821 | 0.767 | 0.700 | 0.814 | 0.742 | 0.650 | 0.807 | 0.608 | 0.560 | 0.643 |
| | | | | acc | sens | spec | acc | sens | spec | acc | sens | spec | acc | sens | spec |
| GC | DMN->AUDI | VAN<-DAN | tRS1_vRS2 | 0.417 | 1.000 | 0.000 | 0.542 | 0.300 | 0.714 | 0.500 | 0.200 | 0.714 | 0.667 | 0.400 | 0.857 |
| | DAN->FPR | VAN<-SN | tRS2_vRS1 | 0.417 | 1.000 | 0.000 | 0.708 | 0.700 | 0.714 | 0.500 | 0.400 | 0.571 | 0.667 | 0.600 | 0.714 |
| | MVISU<-AUDI | FPR<-CB | Loocv_RS1 | 0.417 | 1.000 | 0.000 | 0.417 | 1.000 | 0.000 | 0.625 | 0.600 | 0.643 | 0.708 | 0.600 | 0.786 |
| | MVISU<-FPR | netgc_MVISU->CING | Loocv_RS2 | 0.417 | 1.000 | 0.000 | 0.458 | 0.600 | 0.357 | 0.542 | 0.600 | 0.500 | 0.750 | 0.800 | 0.714 |
| | LVISU<-OVIS | netgc_LVISU->SN | Loocv_conca_RS | 0.417 | 1.000 | 0.000 | 0.729 | 0.700 | 0.750 | 0.563 | 0.500 | 0.607 | 0.604 | 0.550 | 0.643 |
| | LVISU<-FPR | netgc_DMN->AUDI | **Average** | 0.417 | 1.000 | 0.000 | 0.571 | 0.660 | 0.507 | 0.546 | 0.460 | 0.607 | 0.679 | 0.590 | 0.743 |
| | OVIS<-AUDI | netgc_FPL->FPR | | | | | | | | | | | | | |

Grey highlight–accuracy values; yellow highlight–best averages accuracy in each set-up; blue highlight–accuracy > 80%. SVM–support vector machine; LDA—linear discriminant analyses; KNN—K-nearest neighbours; DT–decision tree. Acc–accuracy; sens–sensitivity; spec–specificity; tRS1(2)_vRS2(1)–trained on RS1(or 2) and validated on RS2 (or 1); Loocv–leave one-out cross-validation; conca_RS–concatenation of RS1 and RS2 features.

related in-phase coherence, and static functional connectivity would allow diagnosing the disorder at an overall 81.0% accuracy (average of all machine-learning algorithms tested). Adding CING- and SSM-related EPs and DAN-, AUDI-, and LSM-related GCs, could even improve SVM accuracy up to 95.8%. This shows the following: (i) dynamics metrics performs better than correlation and causal metrics at classifying an ACA in patients with epilepsy; and (ii) combining them with static functional connectivity improves the classifications. This is in line with a recent meta-analysis on the merit of using dynamic functional connectivity to describe psychological outcomes, especially when performance is assessed with objective measures such as IQ tests—as compared to more subjective (self-reported) psychological assessment—[33].

In the clinical context, the default mode, dorsal attention networks, and the salience network being involved and informative to distinguish neuropsychiatric patients from controls is already shown in previous research. In ageing the DMN and DAN are often mentioned and know to be challenged [56–58]. Also the dynamics in terms of causality is shown to correlate

**Table 4. Summary of the best performances of each classifier algorithm: SVM, LDA, KNN, and DT.**

| | List of features | | Validation methods | acc | sens | spec | acc | sens | spec | acc | sens | spec | acc | sens | spec |
|---|---|---|---|---|---|---|---|---|---|---|---|---|---|---|---|
| **EGN + WCoh + sFC** | ip_OVIS-SN_s2 | sFC-DMN-VAN | tRS1_vRS2 | 0.792 | 0.500 | 1.000 | 0.750 | 0.600 | 0.857 | 0.833 | 0.800 | 0.857 | 0.833 | 1.000 | 0.714 |
| | ip_CB-SN_s2 | sFC-MVISU-DMN | tRS2_vRS1 | 0.792 | 0.600 | 0.929 | 0.792 | 0.900 | 0.714 | 0.792 | 0.800 | 0.786 | 0.667 | 0.500 | 0.786 |
| | ip_MVISU-DMN_s2 | In-EP_DMN | Loocv_RS1 | 0.833 | 0.700 | 0.929 | 0.833 | 0.700 | 0.929 | 0.833 | 0.900 | 0.786 | 0.750 | 0.600 | 0.857 |
| | | | Loocv_RS2 | 0.708 | 0.300 | 1.000 | 0.708 | 0.600 | 0.786 | 0.667 | 0.600 | 0.714 | 1.000 | 1.000 | 1.000 |
| | | | Loocv_conca_RS | 0.792 | 0.600 | 0.929 | 0.792 | 0.700 | 0.857 | 0.750 | 0.650 | 0.821 | 0.750 | 0.700 | 0.786 |
| | | | **Average** | 0.783 | 0.540 | 0.957 | 0.775 | 0.700 | 0.829 | 0.775 | 0.750 | 0.793 | 0.800 | 0.760 | 0.829 |
| | | | | acc | sens | spec | acc | sens | spec | acc | sens | spec | acc | sens | spec |
| **EGN + WCoh + GC + sFC** | 82 features = | All best features | tRS1_vRS2 | 0.917 | 0.800 | 1.000 | 0.792 | 0.600 | 0.929 | 0.750 | 0.800 | 0.714 | 0.750 | 0.400 | 1.000 |
| | | without | tRS2_vRS1 | 0.917 | 0.800 | 1.000 | 0.833 | 0.800 | 0.857 | 0.833 | 0.700 | 0.929 | 0.667 | 0.500 | 0.786 |
| | | Out-EP_DMN | Loocv_RS1 | 0.750 | 0.700 | 0.786 | 0.750 | 0.700 | 0.786 | 0.750 | 0.700 | 0.786 | 0.458 | 0.300 | 0.571 |
| | | Out-EP_OVIS | Loocv_RS2 | 0.792 | 0.700 | 0.857 | 0.958 | 0.900 | 1.000 | 0.917 | 0.800 | 1.000 | 0.958 | 1.000 | 0.929 |
| | | Net-EP_DMN | Loocv_conca_RS | 0.938 | 0.850 | 1.000 | 0.833 | 0.750 | 0.893 | 0.792 | 0.700 | 0.857 | 0.542 | 0.450 | 0.607 |
| | | Net-EP_CING | **Average** | 0.863 | 0.770 | 0.929 | 0.833 | 0.750 | 0.893 | 0.808 | 0.740 | 0.857 | 0.675 | 0.530 | 0.779 |
| | | | | acc | sens | spec | acc | sens | spec | acc | sens | spec | acc | sens | spec |
| **EGN + WCoh + GC + sFC** | | | tRS1_vRS2 | 0.750 | 0.600 | 0.857 | 0.792 | 0.600 | 0.929 | 0.750 | 0.700 | 0.786 | 0.750 | 0.400 | 1.000 |
| | | | tRS2_vRS1 | 0.667 | 0.800 | 0.571 | 0.875 | 0.800 | 0.929 | 0.750 | 0.700 | 0.786 | 0.667 | 0.500 | 0.786 |
| | All best 86 features | | Loocv_RS1 | 0.833 | 0.700 | 0.929 | 0.833 | 0.700 | 0.929 | 0.583 | 0.400 | 0.714 | 0.458 | 0.300 | 0.571 |
| | | | Loocv_RS2 | 0.875 | 0.800 | 0.929 | 0.917 | 0.800 | 1.000 | 0.583 | 0.400 | 0.714 | 0.917 | 0.900 | 0.929 |
| | | | Loocv_conca_RS | 0.917 | 0.800 | 1.000 | 0.833 | 0.750 | 0.893 | 0.688 | 0.500 | 0.821 | 0.542 | 0.450 | 0.607 |
| | | | **Average** | 0.808 | 0.740 | 0.857 | 0.850 | 0.730 | 0.936 | 0.671 | 0.540 | 0.764 | 0.667 | 0.510 | 0.779 |
| | | | | acc | sens | spec | acc | sens | spec | acc | sens | spec | acc | sens | spec |
| **EGN + WCoh + GC + sFC** | ip_MVISU-DMN_s2 | sFC-DMN-VAN | tRS1_vRS2 | 0.917 | 0.800 | 1.000 | 0.917 | 0.800 | 1.000 | 0.708 | 0.500 | 0.857 | 0.708 | 0.600 | 0.786 |
| | ip_MVISU-DMN_s4 | sFC-MVISU-DMN | tRS2_vRS1 | 0.917 | 0.800 | 1.000 | 0.583 | 0.500 | 0.643 | 0.708 | 0.600 | 0.786 | 0.667 | 0.600 | 0.714 |
| | GC_NetDeg_DAN | sFC-DMN-LSM | Loocv_RS1 | 0.833 | 0.700 | 0.929 | 0.833 | 0.700 | 0.929 | 0.500 | 0.300 | 0.643 | 0.708 | 0.600 | 0.786 |
| | GC_NetDeg_AUDI | sFC-DMN-CB | Loocv_RS2 | 0.875 | 0.800 | 0.929 | 0.708 | 0.500 | 0.857 | 0.875 | 0.800 | 0.929 | 0.792 | 0.800 | 0.786 |
| | GC_NetDeg_LSM | sFC-DAN-SN | Loocv_conca_RS | 0.958 | 0.900 | 1.000 | 0.875 | 0.850 | 0.893 | 0.729 | 0.550 | 0.857 | 0.792 | 0.800 | 0.786 |
| | | sFC-OVISU-SN | **Average** | 0.900 | 0.800 | 0.971 | 0.783 | 0.670 | 0.864 | 0.704 | 0.550 | 0.814 | 0.733 | 0.680 | 0.771 |
| | In-EP_DMN | In-EP_SSM | | | | | | | | | | | | | |
| | In-EP_DAN | In-EP_CING | | | | | | | | | | | | | |
| | | | | acc | sens | spec | acc | sens | spec | acc | sens | spec | acc | sens | spec |
| **EGN + WCoh + sFC** | ip_CB-SN_s2 | sFC-DMN-VAN | tRS1_vRS2 | 0.875 | 0.700 | 1.000 | 0.917 | 0.800 | 1.000 | 0.875 | 0.700 | 1.000 | 0.833 | 1.000 | 0.714 |
| | ip_MVISU-DMN_s2 | sFC-MVISU-DMN | tRS2_vRS1 | 0.917 | 0.800 | 1.000 | 0.792 | 0.800 | 0.786 | 0.792 | 0.600 | 0.929 | 0.792 | 0.600 | 0.929 |
| | | sFC-DMN-LSM | Loocv_RS1 | 0.750 | 0.500 | 0.929 | 0.750 | 0.500 | 0.929 | 0.625 | 0.700 | 0.571 | 0.792 | 0.700 | 0.857 |
| | In_EP_DMN | sFC-DMN-CB | Loocv_RS2 | 0.875 | 0.800 | 0.929 | 0.708 | 0.500 | 0.857 | 0.792 | 0.700 | 0.857 | 0.667 | 0.500 | 0.786 |
| | In-EP_DAN | sFC-DAN-SN | Loocv_conca_RS | 0.917 | 0.800 | 1.000 | 0.875 | 0.850 | 0.893 | 0.854 | 0.750 | 0.929 | 0.813 | 0.700 | 0.893 |
| | | sFC-OVISU-SN | **Average** | 0.867 | 0.720 | 0.971 | 0.808 | 0.690 | 0.893 | 0.788 | 0.690 | 0.857 | 0.779 | 0.700 | 0.836 |
| | | | overall average = | | | | | | | | | | | | |
| | | | 0.810 | | | | | | | | | | | | |

Grey highlight–accuracy values; yellow highlight–best averages accuracy in each set-up; blue highlight–accuracy > 80%. SVM–support vector machine; LDA—linear discriminant analyses; KNN—K-nearest neighbours; DT–decision tree. Acc–accuracy; sens–sensitivity; spec–specificity; tRS1(2)_vRS2(1)–trained on RS1 (or 2) and validated on RS2(or 1); Loocv–leave one-out cross-validation; conca_RS–concatenation of RS1 and RS2 features.

with psychological performances [28]. Static and dynamic functional connectivity of the DMN and DAN might be explained by the compensatory mechanism observed in aging, as described in the CRUNCH model [59, 60]. This compensatory mechanism is defined as the cognitive reserve in the study of [61] wherein DMN and DAN are also involved. Other studies have

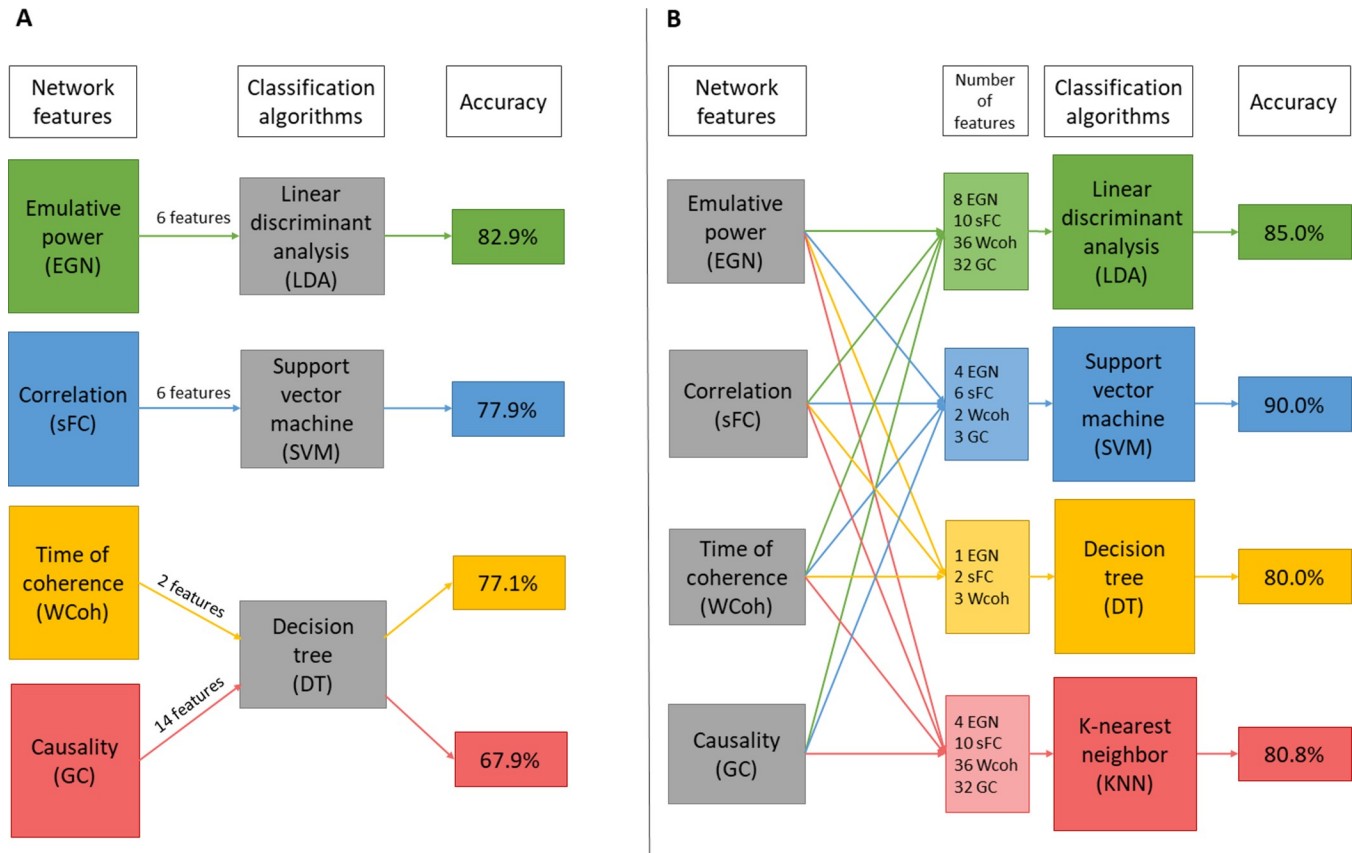

**Fig 6. Graphical summary of the classification results.** A: best performances of each network connectivity metric. B: best performances of each classifier algorithm.

linked cognitive decline with changes in static and dynamic functional connectivity between the DMN and DAN [56, 62, 63]. The salience network (SN), which was part of our sFC best features, and its overall decreased in functional connectivity, is also a marker of brain ageing [57, 59]. SN has also a pivotal role in predicting age and cognitive performances (ranging from episodic memory to executive functions), in healthy elderly with cognitive decline [64]. Regarding mild cognitive impairment, which is a disorder wherein symptoms resemble the deficit in ACA patients, the sFC showed also disturbed network activity in the aforementioned brain regions [27, 65]. Finally in a review paper, from 81 fMRI studies where elderly had a significant cognitive decline, 50% of the studies showed changes in brain activation occurring in the DAN, and 15.3% in the DMN [66].

Sensorimotor networks are also implicated in the process of cognitive ageing [67]. That explains the involvement of the LSM-DMN sFC, the Net GC degree of the LSM, and the In-EP of the SSM, in our best classifier set-up. Finally, the time of in-phase coherence between the pair MVISU-DMN often helped the performances of our classifiers. This might be explained by the anterior-posterior shift observed in ageing in the PASA model of the ageing brain [68]. This shift involved an over-recruitment of the posterior brain functional network (here DMN-MVISU connectivity) in order to compensate for the 'normal' decline in ageing of the visual abilities. And this correlates negatively with the cognitive performances, similar to the increase of in-phase coherence in ACA patients with deterioration in IQ.

ACA patient suffers more specifically from a deterioration in fluid intelligence, which is described as the ability to think logically and solve problems with a limited amount of task-related information [69]. The aforementioned DMN, CING (for cognitive control), and sensorimotor (LSM/SSM) networks have been shown to be linked to this type of intelligence/cognitive processing [70]. The fluid intelligence might come from the ability to switch from perceptive network to processing network, and the ability to switch and hold attention. The DMN, SN and DAN are the 3 networks involved in self-referential processes, dynamic switching between rest and task, and the voluntary allocation of attention [71–73]. Having less copying mechanism here (as shown by the EGN dynamics) in those networks, might reflect a more rigid brain processing, i.e. less fluidic integration, which might explain the poor results in Perceptual Reasoning Index (PRI; fluid intelligence test).

Overall, the DMN, DAN, SN and their dynamic and static functional connectivity to multiple sensory-related networks (visual, sensorimotor) could, no matter the resting-state, help classify correctly patients from healthy subjects. And the involvement of these networks are in line with current research on cognitive decline in ageing, and brain functional connectivity [28, 29, 59, 62, 64, 66, 74, 75].

## 4.2 Technical insights

Evolutionary game theory on networks and its derived emulative powers were the best at describing ACA (highest classifying performances) as compared to the other metrics used in this study. More specifically the In-emulative power of the DMN and DAN were strong biomarkers of ACA. In general, for RS1 and RS2, the DMN In-EPs of the patients were negative, whereas those of the controls, positive. This shows that patients' DMN tends to have a non-emulative (non-replicative) attitude, i.e., all other networks do not tend to copy the DMN activity. For the DAN both groups showed negative in-EPs (non-emulative attitude), but with stronger negative power in ACA patients. The involvement of those networks make sense as seen in the previous paragraph on the clinical relevance. However, the rational for why this novel dynamics metric show stronger discriminatory power as compare to, say, the In or Out-degree of Granger-causality of the DMN and DAN remains unclear. It is indeed difficult to conceive that the replication mechanism between signals do not equate to some degree to a correlation (for sFC) or causal (with GC) effect. In a side study, we actually tried to find similitude between EPs and Deg. GC, which gave us non-significant correlation (between the two metric values). A hypothesis is that, since EGN model is based on differential equations, the emulative values between the networks are influencing not by the network signal directly, but rather by their rate of change in activity ($\frac{dx(t)}{dt}$; where $x(t)$ = RSN activity signal). More details about the EGN model, and its replicator dynamics, in [40].

Regarding SVM and LDA approaches, their ranking in our study is in line with the way these algorithms work. Indeed, in a setup with features having co-variability, and normally distributed, LDA will perform better, with is shown when using many of our most relevant features, and trying to find overall separable (with all of the features) means/centroids. On the contrary SVM, which considers all training data points to find the values (vectors) that help to maximize margin between pools of features, is more robust to outliers—which are clearly present in our small dataset. However, we can expect that with a larger multi-dimensional dataset, the two algorithms (LDA and SVM) would not have substantial differences in performance. In order to increase their performances, a solution might be to combine the two algorithms [76, 77]. The decision tree algorithm is attractive in its simplicity: sequential splitting of the features with thresholds, but might be too dependent on the training sample—as shown by our strong variance in classification accuracy over the 5 validation methods. Regarding KNN, a more

thorough testing with different K values might have been of benefit. Other type of classifier such as elastic net, logistic regression, or naïve Bayes classifier could possibly be applied as well. However, they have shown in general a substantial lower power at classifying neuropsychiatric disorders, as seen in the summary tables of [78].

Finally, for all classifiers presented here, the Matlab default set-ups of each of the machine learning algorithm were used. We only tried, on our best SVM classifiers, to change the kernel, from Gaussian to polynomial; but this had given us less accurate results. More hyper-parameter optimization could be investigated. Also, the worst—but simplest to explain—algorithm, namely decision tree (best at 80%, using only 6 features, with GC), could further be optimized. For instance, by changing node splitting rules from an ascending impurity gain (of Gini's Diversity Index) to descending one; or by changing directly the predictor selections (which features are used in the nodes of the tree) with curvature tests, or interaction tests.

However, since the results are quite robust in SVM, the first step might be to reproduce the results with new (unseen) data, and see if these classifiers continue to provide high performances. Other neuropsychiatric disorders could also be tested with the same pipeline. E.g. autism could be investigated using the rs-fMRI datasets of ABIDE repository (http://fcon_1000.projects.nitrc.org/indi/abide/). With such a large dataset, the reliability of the accuracy obtained using our pipeline is much greater, and the optimisation would be more worthy.

As mentioned in the introduction, a few fMRI research attempt similar classification of neuropsychiatric disorders using more advanced machine learning algorithms, such as convolutional neural networks, or deep auto-encoder [79, 80]. The added value is that those algorithms can be used not only in the supervised multi-labels classification process, but also directly in an unsupervised feature extraction. Though, an important drawback of these advanced classification techniques is that they usually need an important amount of data to be trained upon, which are often not available in case of rs-fMRI data of neuropsychiatric disorders. Also, the understanding and explanation of the feature maps obtained at different CNN layers can be arduous more computationally intensive and troublesome to be clearly explained and descriptive.

Even though research in advance AI for psychoradiology seems promising in many fields, especially segmentation and auto-labelling of cancer or lesion for instance, common machine learning algorithms such as LDA or SVM make the results more understandable and explainable for the clinicians, radiologists and ultimately the patients. For instance, the linear combination of static and dynamic functional connectivity between a handful of well-described networks, can be easily display, and showing how the algorithm split the populations (patients vs controls) is readily feasible. Such simple design and visualization of the process make possible targetable treatment that must be less complicated to design. And following the evolution of such dynamic or static fMRI-based brain features is rather simple, and thus can greatly help to refine classification models to a prognostic level; or to assess lasting effects of different treatments of psychiatric disorders.

## 4.3 Limitations

There are several limitations in this study. First, as mentioned in the previous paragraph, the study would benefit from a larger dataset. Testing the classifiers on new and unseen data extracted from another scanner (different parameters, e.g. TR or duration of the rs-fMRI), and from another site (different medical centre, or country) would also provide insight regarding the robustness of the methods and the features presented here. However, gathering more fMRI data from ACA patients is not an easy task, since this epilepsy co-morbidity is only recently described, and acquiring systematic rs-fMRI data from elderly patients with epilepsy is not

common practise yet. Moreover, having a bigger in-house dataset of participants would not mean getting lower performances, but rather more stable. Indeed, it has been proven by testing pools of data of different sizes, that the accuracy of SVM classification gets simply more reliable less variability in performances as the training sample size increases, in case of data from schizophrenic patients acquired with the same protocol [81]. Nevertheless, the cross-center robustness still needs to be tested.

Second, with high performances of binary classification, we showed a strong diagnostic power of the dynamics features. However, the ACA patients were already diagnosed with ACA based on a clear loss of fluid IQ after psychological assessment. Hence, having an automated tool that diagnoses ACA with lower accuracy than diagnoses based on psychological tests are not needed. In that regards, our goal was not to show the highest performance, but to rank the dynamics features, and show the benefit of combining different dynamic features, using solely rs-fMRI data. Also, with refinement, this analysis could extrapolate from the two class (binary) classification/diagnosis to a multi-class separation, and show prognostic power, for example in order to predict mental ages, or rate of decline in intelligence of patients with ACA (based on a stratified approach, through severity of the ACA for instance).

Despite those limitations, we are confident that the emulative powers, and the times of in-phase coherence of large-scale resting-state networks are informative descriptors of neuropsychiatric disorders, and that combining them with known metrics such as sFC and GC can lead to accurate diagnoses. One could expect similar performance, when testing the new metrics upon schizophrenic, autistic, or depressive patients for example. And this pipeline to train data and validate clinical diagnoses of (neuro)psychiatric disorders can be of great help in clinics and contribute to the field of psychoradiology [82].

## 5 Conclusions

We presented a novel metric for neurodynamics extracted from the evolutionary game theory on network approach: The emulative power of brain networks, i.e. the propensity of a network for replicating another network activity. The emulative powers of the default mode network and dorsal attention network seem defective in ACA; and using these features combined with in-phase coherence measurements, and static functional connectivity yield to an average diagnostic accuracy of 90%, using the SVM algorithm. With regard to the field of neuropsychoradiology, other neuropsychiatric disorders could be tested using similar fMRI-based dynamics features, and could lead to better understanding, and subsequently better treatments of mental disorders. Also, with refinements of the models and the classification pipeline, we could envisage that the fMRI-based neurodynamics presented here would have a prognostic value, for example if labels are not simply binary (stratified models), other feature types are added (multimodal approaches), and using cohort study design (longitudinal cross-sectional studies). This pipeline applied on larger datasets, and if replicated, would demonstrate robustness of these new neurodynamics metrics and reliability of the methods, and hence could be to some extent clinically used.

## Supporting information

**S1 File.**
(PDF)

**S1 Data. Network time series (.mat) and network images.**
(ZIP)

## Author Contributions

**Conceptualization:** Antoine Bernas.

**Data curation:** Antoine Bernas, Lisanne E. M. Breuer.

**Formal analysis:** Antoine Bernas.

**Investigation:** Antoine Bernas, Lisanne E. M. Breuer.

**Methodology:** Antoine Bernas, Svitlana Zinger.

**Project administration:** Albert P. Aldenkamp.

**Supervision:** Albert P. Aldenkamp, Svitlana Zinger.

**Validation:** Albert P. Aldenkamp, Svitlana Zinger.

**Visualization:** Antoine Bernas.

**Writing – original draft:** Antoine Bernas.

**Writing – review & editing:** Albert P. Aldenkamp, Svitlana Zinger.

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
