## [Decision Letter · Decision Letter 0]

7 Dec 2020

PONE-D-20-28406

Emulative, coherent, and causal dynamics between large-scale brain networks are neurobiomarkers of Accelerated Cognitive Ageing in epilepsy.

PLOS ONE

Dear Dr. Antoine Bernas,

Thank you for submitting your manuscript to PLOS ONE. After careful consideration, we feel that it has merit but does not fully meet PLOS ONE’s publication criteria as it currently stands. Therefore, we invite you to submit a revised version of the manuscript that addresses the points raised during the review process.

We look forward to receiving your revised manuscript.

Kind regards,

Pan Lin

Academic Editor

PLOS ONE

Journal Requirements:

Reviewers' comments:

Reviewer's Responses to Questions

**Comments to the Author**

1. Is the manuscript technically sound, and do the data support the conclusions?

Reviewer #1: Yes

Reviewer #2: Yes

2. Has the statistical analysis been performed appropriately and rigorously? 

Reviewer #1: Yes

Reviewer #2: N/A

3. Have the authors made all data underlying the findings in their manuscript fully available?

Reviewer #1: Yes

Reviewer #2: No

4. Is the manuscript presented in an intelligible fashion and written in standard English?

Reviewer #1: No

Reviewer #2: Yes

5. Review Comments to the Author

Reviewer #1: Overall, the study explores the biomarkers from the functional connectivity perspective among ACA in a scientifically sound method. The study has novelty in exploring the interactions between compound networks in the brain particularly in ACA subjects.

There are a few constructive criticisms to help improve the manuscript, which are as follows:

the ABSTRACT section only include one paragraph and don’t use any abbreviations.

Original text: grey highlight = accuracy values; yellow highlight = best averages accuracy in each set-up; blue highlights = accuracy > 80%.

Suggestion: Do not highlights any words other than the numbers in the tables of the text.

Original text: Table 3 and Table 4

Suggest to attach some graphs to depict the results of Table 3 and Table 4 intuitively.

Many abbreviations of terms were used in the manuscript. It is suggested that check them all, ensure they were abbreviated on the first time and use their abbreviations in the rest of manuscript. On the other hand, although it is allowed that using brackets to supplement or explain the main text, but there are too many of them in this manuscript. It is suggested that delete some dispensable contents in brackets or write them in the sentence directly.

Suggest the manuscript should be smoothed in English writing.

Such as “ An important drawback of this classification technique is that it usually needs a large amount of data to be trained upon; plus, the understanding and explanation of the feature maps obtained at different CNN layers can be arduous and troublesome.”, the word ‘plus’ should be replaced by a more formal one.

“Finally, evaluation of brain dynamics can be done through an evolutionary game theory on networks (EGN) approach.”

Suggestion: Finally, evaluation of brain dynamics can be done through an evolutionary game theory (EGN) approach on brain networks.

“Figure 4 depicts the neurodynamics features extraction of the 4 approaches. Those features were extracted for each RS session.”

Suggest to cite the Figure 4 in the main body of the manuscript rather than put it as an independent paragraph. Besides, it is suggested that add serial annotations on different part in Figure 4 and cite them in the main text of 3.2 section.

Reviewer #2: This paper studied the classification between ACA in epilepsy and normal people, using the machine learning method and functional connectivity matric from fMRI. The authors constructed four kinds of functional matric, and then extracted the significant features that were put into the four classifiers to divide the ACA and control subjects. This topic combining brain networks and machine learning methods is the frontier in network neuroscience. This work is well organized, the methods are clear and the results looks like confident. I will recommend the paper published in plos one after a minor revision.

1. This work actually used four kinds of functional matric, i.e., sFC, GC, EGN and WCoh. The sFC, GC and EGN are actually static matric, and WCoh is dynamics. So I suggest that the ‘dynamic features’ (such as below the Figure. 2 description) should be revised to network features or some things. Meanwhile, in the title and abstract (second paragraph), there is no description about sFC.

2. In the abstract, the SVM should be given the full name.

3. Sec. 3.3, NEP is net-EP? I also noticed that the permutations tests were performed to detect the significant difference of network features between two groups. Through the paper, I think that this test should detect the significant features within each group, because the comparison between ACA and control groups is performed at Sec. 4.1.

4. In fig5, ‘s’ is missed for the x-labels, and what does it mean for three ‘+’.

5. Sec4.2, ‘Note that ….. the classification accuracy when added’, can you explain why? The important idea of this paper is to detect significant features that are supposed to be better for the classification, if the not-best features can also improve the accuracy, does it mean that the ‘significant features’ are actually not significant?

6. In table3, I noticed that the term of WCon mixing Loocv_RS2 has 100% accuracy. If this accuracy perfectly classifies the ACA and control subjects, why used another methods and obtained an average accuracy. That is, why using five methods for validation should be explained.

7. Indeed, the GC measures the direct connectivity between regions, the papers ‘Spontaneous electromagnetic induction promotes the formation of economical neuronal network structure via self-organization’ and ‘Investigating driver fatigue versus alertness using the granger causality’ may be helpful for detecting the deeper significant features.

6. PLOS authors have the option to publish the peer review history of their article (what does this mean?). If published, this will include your full peer review and any attached files.

Reviewer #1: No

Reviewer #2: No

---

## [Author Response · Author response to Decision Letter 0]

5 Mar 2021

Dear Editor, dear Reviewers,

We would like to thank you for the interest you have shown in our manuscript, and for your insightful and constructive comments. We have carefully considered and addressed each of the comments.

Please find the detailed answers to your comments in the end of this PDF or in the 'Response_to_Reviewers.docx' file.

Sincerely,

Antoine and all co-authors

---

## [Decision Letter · Decision Letter 1]

5 Apr 2021

Emulative, coherent, and causal dynamics between large-scale brain networks are neurobiomarkers of Accelerated Cognitive Ageing in epilepsy.

PONE-D-20-28406R1

Dear Dr. Antoine Bernas,

We’re pleased to inform you that your manuscript has been judged scientifically suitable for publication and will be formally accepted for publication once it meets all outstanding technical requirements.

Kind regards,

Pan Lin

Academic Editor

PLOS ONE

Additional Editor Comments (optional):

Reviewers' comments:

Reviewer's Responses to Questions

**Comments to the Author**

1. If the authors have adequately addressed your comments raised in a previous round of review and you feel that this manuscript is now acceptable for publication, you may indicate that here to bypass the “Comments to the Author” section, enter your conflict of interest statement in the “Confidential to Editor” section, and submit your "Accept" recommendation.

Reviewer #1: All comments have been addressed

Reviewer #2: All comments have been addressed

2. Is the manuscript technically sound, and do the data support the conclusions?

Reviewer #1: Yes

Reviewer #2: Yes

3. Has the statistical analysis been performed appropriately and rigorously? 

Reviewer #1: Yes

Reviewer #2: Yes

4. Have the authors made all data underlying the findings in their manuscript fully available?

Reviewer #1: Yes

Reviewer #2: No

5. Is the manuscript presented in an intelligible fashion and written in standard English?

Reviewer #1: Yes

Reviewer #2: Yes

6. Review Comments to the Author

Reviewer #1: (No Response)

Reviewer #2: Since adding insignificant measures increases the classification accuracy, I guess that the measures must be pooled with a nonlinear manner in classifiers. The author should think deeper about this question and try to find some literatures to support this.

7. PLOS authors have the option to publish the peer review history of their article (what does this mean?). If published, this will include your full peer review and any attached files.

Reviewer #1: No

Reviewer #2: No

---

## [Editor Report · Acceptance letter]

7 Apr 2021

PONE-D-20-28406R1 

Emulative, coherent, and causal dynamics between large-scale brain networks are neurobiomarkers of Accelerated Cognitive Ageing in epilepsy. 

Dear Dr. Bernas:

I'm pleased to inform you that your manuscript has been deemed suitable for publication in PLOS ONE. Congratulations! Your manuscript is now with our production department. 

Kind regards, 

on behalf of

Dr. Pan Lin 

Academic Editor

PLOS ONE